# Can mutual health organisations influence the quality and the affordability of healthcare provision? The case of the Democratic Republic of Congo

Bart Criel[1]*, Maria-Pia Waelkens[2], Fulbert Kwilu Nappa[3], Yves Coppieters[4], Samia Laokri[5]

1 Department of Public Health, Institute of Tropical Medicine, Antwerp, Belgium, 2 School of Public Health, Université Libre de Bruxelles (ULB), Brussels, Belgium, 3 Department of Health System Management, Kinshasa School of Public Health, University of Kinshasa, Kinshasa, Democratic Republic of Congo, 4 School of Public Health, Health Policy and Systems–International Health, Université Libre de Bruxelles, Brussels, Belgium, 5 School of Public Health and Tropical Medicine, Global Community Health and Behavioral Sciences, Tulane University, New Orleans, LA, United States of America

* bcriel@itg.be

## Abstract

### Background

In their mission to achieve better access to quality healthcare services, mutual health organisations (MHOs) are not limited to providing health insurance. As democratically controlled member organisations, MHOs aim to make people's voices heard. At national level, they seek involvement in the design of social protection policies; at local level, they seek to improve responsiveness of healthcare services to members' needs and expectations.

### Methods

In this qualitative study, we investigated whether MHOs in the Democratic Republic of Congo (DRC) succeed in defending members' rights by improving healthcare quality while minimising expenses. The data originate from an earlier in-depth investigation conducted in the DRC in 2016 of the performance of 13 MHOs. We re-analysed this existing dataset and more specifically investigated actions that the MHOs undertook to improve quality and affordability of healthcare provision for their members, using a framework for analysis based on Hirschman's exit-voice theory. This framework distinguishes four mechanisms for MHO members to use in influencing providers: (1) 'exit' or 'voting with the feet'; (2) 'co-producing a long voice route' or imposing rules through strategic purchasing; (3) 'guarding over the long voice route of accountability' or pressuring authorities to regulate and enforce regulations; and (4) 'strengthening the short voice route' by transforming the power imbalance at the provider–patient interface.

**Data Availability Statement:** All relevant data are within the manuscript and its Supporting Information files.

**Funding:** This study is based on material from previous research, commissioned by POMUCO in 2016, and supported by the Belgian Directorate-general Development Cooperation and Humanitarian Aid in the context of the programme MASMUT (Micro-assurance santé/Mutuelles de santé).

**Competing interests:** The authors have declared that no competing interests exist

## Results

All studied MHOs used these four mechanisms to improve healthcare provision. Most healthcare providers, however, did not recognise their authority to do so. In the DRC, controlling quality and affordability of healthcare is firmly seen as a role for the health authorities, but the authorities only marginally take up this role. Under current circumstances, the power of MHOs in the DRC to enhance quality and affordability of healthcare is weak.

## Conclusion

On their own, mutual health organisations in the DRC do not have sufficient power to influence the practices of healthcare providers. Greater responsiveness of the health services to MHO members requires cooperation of all actors involved in healthcare delivery to create an enabling environment where voices defending people's rights are heard.

## Background

The national health policy of the Democratic Republic of Congo (DRC), defined in 1984, is based on the Primary Health Care strategy, as initially defined in Alma Ata in 1978 [1] and later renewed in Astana in 2018 [2]. The central place of Primary Health Care in the Congolese health system is explicitly mentioned in the *Stratégie de Renforcement du Système de Santé* (SRSS) elaborated in 2010 by the Ministry of Health [3]. The health system is organised into three levels: (1) the central level (Ministry of Health), with a normative and regulatory role; (2) the intermediate level with two main structures: a) the Provincial Health Inspections (IPS) with a role of control, audit and inspection, and b) the Provincial Health Divisions (DPS), with a role of coordinating health interventions, planning and technical support; and (3) the operational level, formed by the health zones. The health zones consist of a network of health centres and referral hospital and correspond to what are commonly called 'health districts' in Anglophone African countries. Among the six strategic priorities proposed in the SRSS, the revitalisation of the health zone is ranked as first. The health zone is put forward as the key administrative and operational entity in the implementation of Primary Health Care in the Congolese health system. The focus is on person-centred and integrated health services, with due attention for community participation and for social determinants of health. The 2018 Law on the fundamental principles pertaining to the organisation of public health [4] states in its article 9 that the peripheral level of the health system–i.e. the health zone–has as mission to implement the strategy of Primary Health Care. Private faith-based healthcare providers, of which the largest is the network coordinated by the Catholic Church (BDOM—*Bureau Diocésain des Oeuvres Médicales*), play an important role in healthcare delivery and are in most instances well integrated into the public health system. The health sector suffers from substantial domestic underfunding and heavy reliance on aid [5]. External funding allocation is largely assigned to disease-specific programmes such as the Global Fund to Fight AIDS, Tuberculosis and Malaria, or to a diversity of local projects directly funded by multiple external agencies [6]. Underfunded health authorities progressively lost capacity in leadership and coordination. User fees have become the main source of income for healthcare providers, covering healthcare activities and staff remuneration [7]. This situation induced substantial overcharging and over-prescribing of medicines, diagnostic tests, and medical procedures [8].

On the path towards universal health coverage, the DRC opted for a social protection system based on health insurance, in which mutual health organisations (MHOs; *mutuelles de santé* in French) have a predominant role. The law determining the fundamental principles of MHOs (*Loi organique* N˚ 17/002) was promulgated on 8 February 2017 [9]. It defined an MHO as a non-profit association of members that seeks, via member contributions, to conduct interventions of protection, solidarity, and mutual assistance for its members and their dependants (article 4.4), offering people the opportunity to access quality healthcare at decent prices. The law provides two options: (1) compulsory affiliation for anyone whose premium can be deducted at the source in enterprise-based, corporate, school, and student MHOs; and (2) voluntary enrolment in community-based MHOs for informal sector workers (article 70). This definition highlights the difference between the concepts of 'community-based health insurance' (CBHI) and 'mutual health organisations' (MHOs). CBHI usually refers to health insurance schemes designed for informal sector workers. Affiliation is usually voluntary. The term MHO refers in the first place to the non-governmental nature of the insurance scheme. Members participate in decision-making through representation in the General Assembly. MHOs can be designed for formal and informal sector. Affiliation can be voluntary or compulsory.

As described in more detail in the Background Paper on Community Health Insurance (CHI) established for the 2010 World Health Report, the first MHOs emerged in the DRC in the 1980s [10]. Scheme-specific data are rare. One notable exception is that of the Bwamanda scheme, an African pioneer in community-based health insurance active since 1986. In the following decades, a diversity of social protection initiatives proliferated, and the country experienced a rapid expansion of MHOs [11]. Part of this occurrence can be seen as a reaction to a failing health system. To support this dynamic, a National Programme for the Promotion of MHOs (PNPMS—*Programme national de promotion des mutuelles de santé*) was created in 2001. Online newspapers incessantly report on events, anniversaries or inaugurations of MHOs (see for instance the newspaper ouraganfm.com but also the professional websites http://pomuco.org/info.php and http://www.cgatrdc.com/). National population coverage remains low at 1.2% [12], but with higher coverages in individual schemes. Emerging MHOs often attract few members, face unforeseen challenges and disappear soon after their creation [13]. In 2009, the PNPMS designed a new plan to organise support for the MHO movement and professionnalise its management. Technical support centres were created to accompany MHOs. In 2015, new and pre-existing support centres (Table 1) formed a national platform: POMUCO (*Plateforme des organisations promotrices des mutuelles de santé du Congo*).

In 2016, POMUCO commissioned a study of the performance of MHOs in the DRC and their potential role in advancing towards universal health coverage [14]. A key concern of POMUCO was to have more insight in the ability of Congolese MHOs to improve the quality and the affordability of healthcare services provided by the contracted healthcare providers. Three of the authors of the present paper (BC, MPW, FNK) were very closely involved in the POMUCO study. It is on these 2016 data that the current study is based.

The providers' tendency to seek maximal payment from patients challenges MHOs in their mission to achieve better access to quality healthcare services [15]. The MHO role is therefore not limited to providing insurance–financial coverage of health services when needed in return for a prepaid premium–but also comprises interventions to reduce expenses for healthcare services while maintaining quality standards. Conceptual frameworks describing the potential role of MHOs in improving healthcare delivery distinguish several levers MHOs can use to assert their influence:

- Financial lever: providers will seek increased, or at least regular, income, which at the same time incites and enables them to improve care quality [16].

**Table 1. The technical support centres forming POMUCO.**

| | |
|---|---|
| CAMS | "Cellule d'appui aux mutuelles de santé" Created in 1997, CAMS supports a network of 23 MHOs: REMUSACO (*Réseau des mutuelles de santé communautaires*) |
| CENADEP | "Centre national d'appui au développement et à la participation populaire" Founded in 2000, CENADEP's main concern is to assist communities in realising their own projects. Their involvement in MHOs is recent. |
| CGAT | "Centre de gestion des risques et d'accompagnement technique" Founded in 2010, the CGAT has set up technical support centres in the provinces of Congo-Central, Kinshasa, Equateur and North Kivu. |
| UMUSAC | "Union des mutuelles de santé du Congo" UMUSAC is the branch of the Christian labour movement of Congo (le Mouvement ouvrier chrétien du Congo (MOCC)) dedicated to the support of MHOs. It was restructured in 2009 and currently coordinates networks of MHOs in the provinces of Haut-Katanga, Kinshasa, Kwilu, and Tshopo. |
| PRODDES | "Le Réseau pour la promotion de la démocratie et des droits économiques et sociaux" Founded in 2008, PRODDES is concerned with empowering civil society organisations. Its member organisation CRAFOP (Comité de Réveil et d'Accompagnement des Forces Paysannes) supports MHO development in the province of Equateur. |
| CDI Bwamanda | CDI Bwamanda supports 3 MHOs in the province of South Ubangi. It became member of POMUCO in 2017. |

- Member activism: as a collective of users, MHOs can put pressure on providers to better meet their demand [17].

- Strategic purchasing: MHOs can actively search for the best providers, and negotiate, in a context of competition, quality, prices, and payment methods [18].

- A contractual relationship: the negotiated conditions are laid down in a contract that describes services to provide, quality standards to be observed, tariffs and terms of payment, and a mechanism to enforce the contract terms [19].

These principles shaped the design of DRC MHOs. Their organisational structure as member organisations is meant to defend people's choices and make their voices heard. Contracts describe the conditions that healthcare providers must meet to receive payment. These conditions include six measures to promote quality healthcare and cost containment:

- Availability of essential medicines.

- Observance of the list of services covered by the MHO.

- Gatekeeping: Hospital care is covered only for patients referred by a first-line health facility.

- Observance of agreed tariffs: Tariffs are subject to negotiation between MHO and provider. The contracts mention both tariffs charged to the MHO and co-payment charged to the patient. Co-payment for basic services and generic medicines is often lower than for expensive alternatives. The underlying assumption is that providers will consider cost-effective prescribing to avoid high expenses for their patient.

- Use of essential generic medicines, with conditions for use of non-generic medicines.

- Respect for national guidelines for diagnosis, treatment and preventive care, to promote rational prescribing, ensuring quality at the lowest possible cost.

In the context of the DRC, we aimed at assessing the effectiveness of MHOs on improving access to affordable and acceptable quality care and stating whether or not they constitute a powerful mechanism to move towards universal coverage. This line of enquiry is consistent with the emphasis on empowerment and transformation in the framing and assessment of

Community Health Insurance (see ref WHR 2010). Therefore, we analysed power imbalances that can mitigate existing social vulnerability of consumers using a comprehensive framework that has been developed in the Indian context by Michielsen et al (2011) [20].

## Methods

Our qualitative study of the capacity of MHOs to influence quality and affordability of healthcare provision builds on a wider investigation of the performance of MHOs in the DRC, as commissioned by POMUCO in 2016 and conducted in the period May-October of that same year. The results of the this investigation suggested that the failure of MHOs to change healthcare provider practices was a major obstacle to improve performance. At that time, we did not further explore the reasons for this finding, but decided in 2019 to re-analyse these data, focussing on this particular issue, using framework analysis [21] and more particularly the framework depicted in Fig 1 and adapted to the context of Congolese MHOs (see Table 2).

In the POMUCO study, structured in-depth interview was the main tool for data collection, complemented with data collection from annual reports and routine monitoring. In-depth interviews were conducted in 13 MHOs in the 4 provinces of Kinshasa, Kwilu, North Kivu, and South Kivu. This sample was defined to provide the needed information while remaining manageable. We selected 10 MHOs that were functional and 3 that had failed. Other selection criteria were exploration of a variety of contexts (geographical, economic, urban/rural, longevity of the MHO, the type of technical support received, etc.) and accessibility of the sites. For each functional MHO, we interviewed the managers, members of the board of directors, selected healthcare providers, health zone officers, technical support centres, and additional key informants like local health authorities and leaders.

Interview guides were developed for each type of respondent. The guide for MHO managers and leaders was the most comprehensive. It was composed of 9 sections covering the creation of the MHO, its governance, management, membership, resource mobilisation, healthcare services, MHO–provider relationship, support, and results. Interviews of additional key informants probed opinions about the performance of MHOs, the support they received and their social effects. Each interview guide focused on theory (intention) and practice (implementation).

Data collection was carried out in June and August 2016 by three teams of one main and one assistant researcher. In total 64 interviews were conducted: 22 in North-Kivu, 17 in South Kivu, 20 in Kinshasa and 5 in Kwilu. Interviews with MHO managers and leaders were in most cases group interviews spread over 2 days. Other interviews took on average 60 minutes.

The persons interviewed were either MHO managers, health professionals, or representatives from local health authorities who were well aware of the study. Patients or individual MHO members were not interviewed. The data collected did never pertain to any sort of personal information. Interviewees were well aware of the fact that the study had been formally commissioned by POMUCO. Permission to record and use the transcribed information anonymously was obtained orally, after discussing the purpose of the interview, voluntary participation, and the option to interrupt the interview at any time. This information was repeated to any respondent joining the group interview at a later stage: e.g. members of the MHOs board or hospital-based medical doctors. One person refused the interview to be recorded, but agreed that written notes could be used. Several interviewees requested to interrupt the recording during a short period to share information they considered to be sensitive. This information was not used in the data analysis but contributed to our in-depth understanding of the overall context. Finally, the interviewed persons, or their representatives, participated in a three-day workshop where the study report was reviewed and eventually approved.

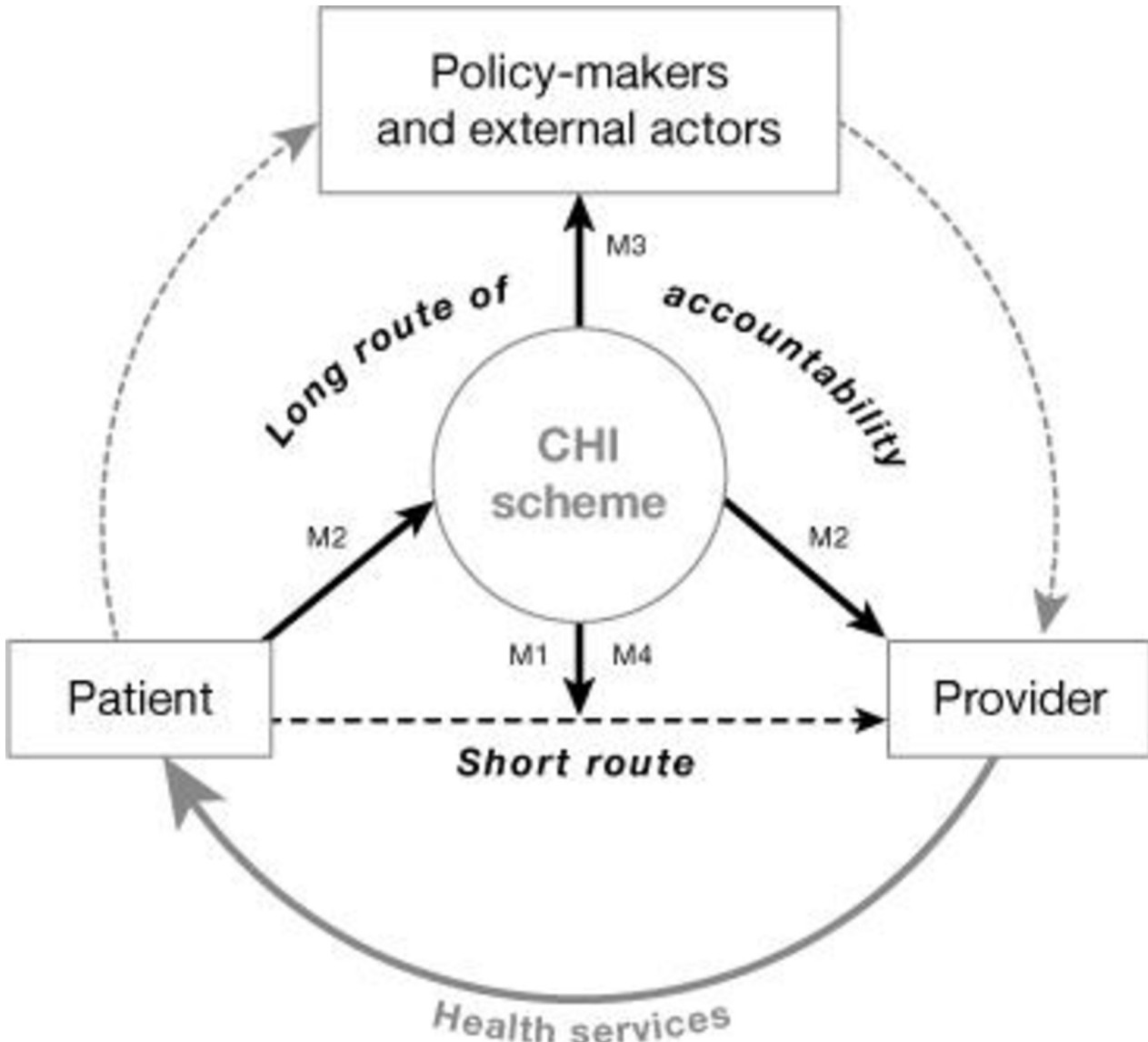

CHI refers to community-based health insurance, that includes other models than Mutual Health Organisations, the prominent model in African countries.

**Fig 1. Explanatory framework of routes to influence care provision.**

Interviews conducted in French were transcribed verbally by the research assistants. Transcriptions were counterchecked by the main researcher. For interviews conducted in the local language (Kikongo in the Kwilu province), audio recordings were summarised by the team who conducted the interviews.

We built on the original framework of Michielsen [20] that distinguishes four mechanisms (routes) by which MHOs may improve the provision of quality services (Fig 1):

**Table 2. Analytical framework.**

| | |
|---|---|
| **M1—Exit route: freedom of choice to access quality providers and exit low-performing providers** | |
| Measure; outcomes | First selection of providers |
| Measure; outcomes | Members' choice among providers |
| Outcome | Exit from the MHO |
| **M2—Co-producing a long voice route: imposing rules through strategic purchasing** | |
| Measure; outcomes | Quality assessment before inclusion of providers |
| Measure | Affordability assessment |
| Outcome | Balance premium–healthcare expenses |
| Measure; outcomes | Contract |
| Measure; outcomes | Availability of essential medicines |
| Measure; outcomes | Respect of services covered |
| Measure; outcomes | Gatekeeping for hospital care |
| Measure; outcomes | Respect for agreed tariffs |
| Measure; outcomes | Use of generic and non-generic medicines |
| Measure; outcomes | Respect of (national) treatment guidelines |
| Measure; outcomes | Control by medical advisor |
| Outcomes; conditions | Contribution of MHO to quality? |
| Outcomes; conditions | Effectiveness of cost-containment measures overall |
| **M3—Guarding over the long voice route: link with more voiced groups or hold authorities accountable to regulate health services** | |
| Measure; outcomes | Initiatives for cooperation |
| Measure; outcomes | Support organisation |
| Measure; outcomes | Platform for dialogue |
| **M4—Short voice route: transforming the power imbalance at the provider–patient interface** | |
| Measure; outcomes | Presence of MHO delegate in health facilities |
| Measure; outcomes | Information of members |
| Measure; outcomes | Feedback mechanism members -> MHO -> providers |
| Measure; outcomes | Members' control over decision-making in the scheme |
| Outcomes; conditions | Capacity of the MHO, as an association of members, to promote change |

1. The exit route (M1): 'voting with the feet'; freedom of choice to access quality providers and exit low-performing providers.

2. Co-producing a long voice route (M2): strategically purchase healthcare from providers, which gives a mandate for setting quality standards.

3. Guarding over the long route of accountability (M3): MHOs link communities with politically more voiced groups or hold government accountable to regulate the health system.

4. Short voice route (M4): transforming the power imbalance at the provider–patient interface; social and emancipatory programmes increase peoples' confidence to discuss directly with providers.

For each of these mechanisms, the authors describe possible measures the MHOs can implement, contextual requirements, and expected outcomes (S1 File).

Information on healthcare delivery and MHO ability to influence service provision was extracted and classified manually according to a pre-established framework. Table 2 shows the analytical framework we adapted from Michielsen et al (2011) [20] to the operational context. We listed all measures to influence healthcare provision put in place by the MHOs under the appropriate route (M1 -> M4). For classifying data, we used the sequence: the measure (design

and implementation), its outcome, and explanations given by respondents about the measure's effectiveness. We omitted 'contextual requirements' because determining contextual requirements was part of the analysis. Instead, in order to fit the purpose of our study, we re-grouped opinions and explanations concerning three major themes in our investigation, i.e. the capacity of the MHO to promote quality, cost-containment and change. Ex-post the data analysis, a number of adaptations to the framework are proposed and are explained in the discussion section.

The analysis focused on whether the measures introduced by the MHOs achieved their purpose, why or why not, and which circumstances would be required for better results. For each measure, we looked at the progression from (1) objective → (2) design (procedures to reach the objective) → (3) implementation → (4) result, considering internal and external factors that influenced each step.

This analysis was first done per MHO, giving a diagnosis for each scheme. Comparison between the different MHOs gave insight into the reasons why some MHOs performed better than others.

Thereafter, merged data of all MHOs was classified and analysed per measure. This alternative comparison highlighted mainstream tendencies and allowed additional insights regarding why outliers performed differently. Key factors and results feeding this analysis are available online (S2 File). Results were triangulated with those of the overall investigation of the performance of the MHOs that had prompted a second analysis of one of its crucial findings.

The study protocol was approved by the National Ethics Committee of the DRC Ministry of public health. Informed consent obtained from respondents included assurance of anonymous reporting of information. To protect respondent confidentiality, quotes in the results section are not marked by identifiers. However, the context indicates the relevant respondent characteristics, either the MHO in question or the type of respondent.

## Results

Of the 64 interviews, 57 provided relevant information on healthcare provision, 15 of them with MHO managers and/or members of the board, 18 with providers (health centre, medical centre, hospital, network coordinator), 12 with members of support centres, 10 with health authorities, and 2 with NGOs.

Of the 13 MHOs, nine were retained for analysis. Interviews with the managers of the three MHOs that had failed were excluded due to insufficient detail on healthcare delivery. For one functional MHO, the health staff themselves constituted the MHO membership. Consequently, the influence of the MHO on care provision had not been discussed. Fig 2 and Table 3 give an overview of the nine retained MHOs.

MUSECCO, formerly MUSEKIN, was one of the oldest MHOs in the DRC. Its membership, mainly formed by teachers of the catholic network, declined since the creation of the Kinshasa branch of MESP in 2011, a national MHO for school teachers that benefited from government subsidies, and membership contributions directly deducted from salaries. Lisanga, KLA and MUSSRA were designed according to a model developed by the CGAT (Centre de Gestion des risques et d'accompagnement technique) in 2010. UMUSAC is categorised simultaneously as MHO and as support centre. The UMUSAC coordination team provided technical support to other MHOs in Kwilu but ensured central management of the 12 MHOs of its own network. In our study, this network was considered a single MHO. MUSOSA was, after some difficult years, revising its strategies at the time of the study. Nyantende and Walungu were respectively among the first and most recently created MHOs of the South Kivu network.

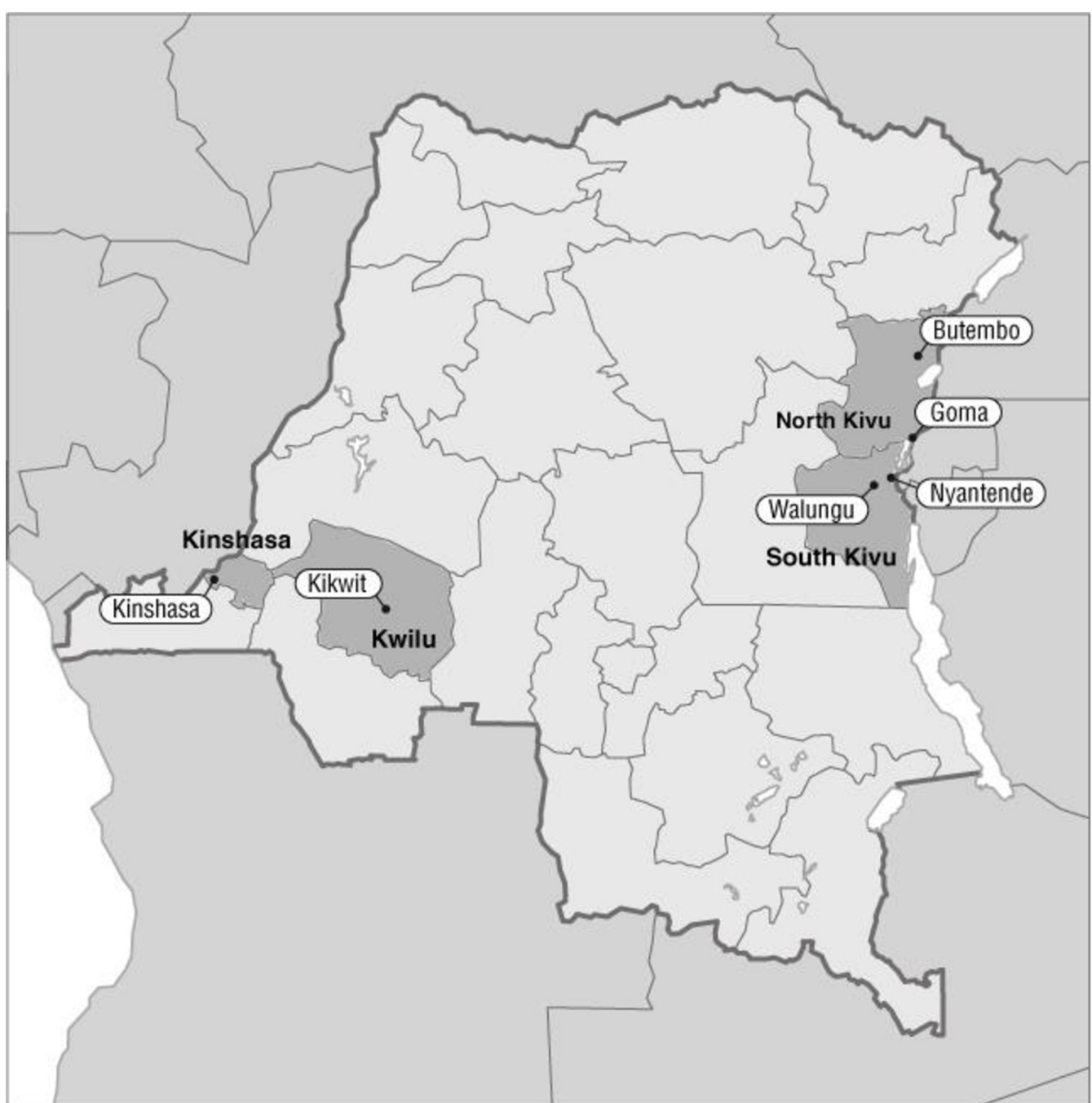

**Fig 2. Provinces and locations of investigation.**

The degree of professionalization of MHO management ranged from the highly qualified and salaried staff of the MESP to the volunteers of the MHOs of the UMUSAC network, receiving a modest allowance. Before inauguration, all MHOs had a period of intensive preparation for which they received assistance from specialised national or international organizations. This preparation consisted of training of MHO leaders and managers, a large-scale awareness campaign and a structured technical preparation.

**Table 3. MHOs included in the study.**

| MHO | Province | Location | Support centre | Inauguration | Beneficiaries 2015 | Annual premium/ beneficiary 2015 ($) |
|---|---|---|---|---|---|---|
| **LISANGA** | Kinshasa | Kinshasa | CGAT Kinshasa | 2011 | 2219 | 54 |
| **MUSECCO** <br> Mutuelle de santé des enseignants des écoles catholiques du Congo | Kinshasa | Kinshasa | (CGAT Kinshasa) [§] | 2000 | 9000 | 36 |
| **MESP** <br> Mutuelle de santé des enseignants de l'enseignement primaire, secondaire et professionnel | Kinshasa | Kinshasa | - | 2011 | 193000* | 26 ** |
| **UMUSAC** <br> Union des mutuelles de santé du Congo | Kwilu | Kikwit | UMUSAC/ MOCC | 2010 | 44922 | 12 *** |
| **Kingo la Afya (KLA)** | North Kivu | Goma | CGAT Goma | 2014 | 3119 | 25 |
| **MUSSRA** <br> Mutuelle de santé Saint-Raphaël | North Kivu | Goma | CGAT Goma | 2015 | 2145 | 25 |
| **MUSOSA** <br> Mutuelle de solidarité pour la santé | North Kivu | Butembo | (CGAT Beni) [§] | 2010 | 2619 | 10 |
| **Nyantende** | South Kivu | Nyantende | CAMS | 2001 | 12474 | 5 |
| **Walungu** | South Kivu | Walungu | CAMS | 2009 | 2738 | 5 |

[§] Loose connection to the CGAT network; technical support when needed.

* in 2016.

**calculated: 120 $ per household head; average household size = 4.7.

***estimated: 12,000 Francs congolais (Fc) for adults, 6000 Fc per child.

The results of the framework analysis are summarised and organised under the four routes by which MHOs may improve healthcare provision.

## M1—exit route

The assumption underlying the influence of the exit route is that poor-performing providers will improve care quality and affordability to prevent insured patients from going elsewhere. In the studied MHOs, the exit route did not instigate this expected outcome.

We distinguished exit by individual members and by MHOs in their process of selecting providers.

The power of individual exit was intrinsically restricted in five MHOs that required their members to register in a health facility close to their home, namely MESP and MUSECCO in Kinshasa, KLA and MUSSRA in Goma, and UMUSAC in Kikwit. In South Kivu, members were allowed to attend any of the health facilities with whom one of the 23 existing MHOs has established a contract. In practice geographical access limited freedom of choice. Only members of MUSOSA in Butembo and Lisanga in Kinshasa, had unrestricted choice among contracted providers. In Kinshasa, however, members complained that "elsewhere will not be better, financial exploitation of patients is the rule". All respondents involved in MHO management reported member complaints about health staff attitudes and high out-of-pocket expenditure, often because of over-prescribing and direct charging of services that the MHO did not cover. To voice their frustrations, members did 'vote with their feet', but without affecting healthcare providers: they left the MHO.

Selection of providers was not an option for MHOs in rural areas. MHOs worked with the referral hospital and the network of health centres of the public sector. In urban areas, medical advisers considered selection of providers the most straightforward way to offer the best available quality/price combinations to their members. However for most, the threat of exit was not an argument in their arrangements with providers, because member numbers were small. The exception was MESP, which had a large membership. The coordination of the Catholic network of health facilities (BDOM) accepted the conditions imposed by MESP "because if we refuse, we lose a large population". Yet this understanding at coordination level did not trickle down to the level of their health facilities where providers did not perceive the same pressure to improve care quality and affordability.

## M2—co-producing a long voice route: Imposing rules through strategic purchasing

Three key mechanisms used by the MHOs to strategically purchase healthcare were 1) a quality and affordability assessment of potential providers, 2) a contract defining conditions for care provision and payment, and 3) control of its application by a medical adviser. Although procedures were well designed, so far their outcomes did not meet expectations. Where quality and affordability assessments were carried out, the restricted choice of providers that met the requirements limited their usefulness for selecting providers. The rules laid down in the contracts had little effect on provider practices. The control of their application was hampered by the reluctance of providers to accept the authority of the medical advisors.

Firstly, quality and affordability assessment of potential providers was seen by the medical advisors as an essential step to prepare a sustainable relationship with providers. With respect to quality, six out of nine MHOs, namely MESP, MUSOSA, MUSECCO and the three MHOs of the CGAT network, carried out a formal assessment. The CGAT, for example, had developed an evaluation tool with criteria for human resources, hygiene, patient care in outpatient and inpatient departments, maternal care, availability and management of medicines, laboratory and operation theatre. The two South Kivu MHOs and UMUSAC simply included the health facilities accredited by the health zone. With respect to affordability, for health facilities that achieved the required quality standards, the CGAT proceeded with a financial feasibility study that provided the basis for negotiating tariffs and setting member premiums. In practice, health facilities with insufficient quality scores still were accredited because the high fees charged by quality providers would have resulted in too-expensive premiums. The medical advisers set themselves the task to work with accredited providers on quality improvement and cost-containment. In contrast, the MHOs of South Kivu fixed premiums based on people's disposable income without calculating expected expenses. They offered a comprehensive service package that exceeded member contributions. This income/expenses imbalance had severe consequences, not only for the survival of the MHOs but also for their relationship with providers. All respondents in South Kivu described the untenable situation of MHOs that had depleted their funds before the year's end, some after fewer than 6 months, yet continued to issue proof of entitlement to members needing healthcare–"after all, they had paid their premium". In 2015, only 3 of the 23 MHOs that form the South Kivu network could fully honour their invoices for healthcare with members' annual contributions [22]. As a result, "more and more providers refuse any further dealings with insolvent MHOs". Meanwhile, MUSOSA in North Kivu was carrying out a new financial feasibility study with the assistance of the health authorities of the District level (now called Health Branch of Butembo (*antenne sanitaire de Butembo*). This joined exercise had a positive spinoff: It made the authorities "better understand the necessity of imposing generic medicines and rational prescribing" to reduce expenses.

Secondly, all MHOs had established contracts with their healthcare providers that described the terms of cooperation, the routine administrative procedures, and conditions for care provision and payment. Both MHOs and providers considered the contract an important working tool for discussing disagreements over treatments and invoices. Nevertheless, respondent statements show fundamental differences in appreciating the issues of quality and cost of healthcare that get in the way of effective cooperation. In the opinion of providers, "the MHO leaders think that hospitals want to get rich on the back of the MHO, but the tariffs they propose are too low to provide quality care"; "user fees are our only revenue". The view from MHO side is that "because health facilities are self-financing, they take the MHO for a cash cow"; "we try through discussion and exchange, but we have too little weight to influence care quality, tariffs and rational prescribing". The mismatch of opposing views and objectives, in the context of underfunded healthcare services, affects each of the six measures stipulated in the contract to improve care quality and affordability.

All contracts included a clause imposing on-site availability of essential medicines, which was in their members view the most important factor determining healthcare quality. In practice, the regular income from MHOs made a real difference for providers with a large MHO clientele. Indeed, these healthcare facilities could "replenish stocks as soon as an MHO paid its bill". In the majority of healthcare facilities, however, stock-outs remained frequent despite the clause on drug availability in the contract.

The clause on observance of the list of services covered by the MHO aimed at containing healthcare expenses. All MHOs offered a fairly extensive package of services including the basic first- and second-line healthcare routinely provided in the public health system. In addition, several contracts specified that non-basic services could be covered in certain circumstances. In the MHOs of the CGAT network and MESP, non-basic services could be granted with the approval of the medical advisers. Such approval was not required in the MHOs in South Kivu where granted non-basic services were listed in the contract along with their tariffs. In practice, as reported by nearly all respondents involved in MHO management, "providers tended to deliver and/or invoice services that were not covered or even not needed, because they sought to maximise their revenue". At the time of payment, the list of covered services was essential to amend the invoices and determine the amount the MHO agreed to pay. However, this had "little effect on prescribing": "non-covered services were simply charged to the patient". The reluctance of providers to take account of the list of covered services was illustrated in one urban hospital where physicians preferred not to know who were MHO members and who were not. They said that patients' insurance status should not influence prescribing because this was "discrimination and members should not be penalised". In this case, the contract cannot promote rationalisation of care, and patients end up paying out-of-pocket for services not covered by their coverage plan.

The gatekeeping rule, stipulating that second-line hospital services were covered only if the patient was referred by a first-line health facility, was mainly respected in rural areas. This rule was less effective in urban areas, where, instead of basic health centres, several medical centres were contracted for first-line care. Most medical centres offered on-site an extensive range of specialised medical and surgical procedures. The MESP adapted the referral rule by focusing on the specialised services rather than on physical referral. To control the consumption of medical imaging, specialised examinations, or care, providers had to ask permission from the medical adviser before prescribing them. The medical adviser also decided whether the patient should be referred to a specialised clinic (e.g., for medical imaging, ear-nose-throat, physiotherapy, mental health), even if the required service was available in the health facility where the patient was seen. The MESP deemed this strategy highly effective, since it "reduced expenses by avoiding unnecessary consumption" and "improved the quality through selection

of specialised centres". On the other hand, some providers questioned "the quality of patient follow-up" and "the erosion of the pyramidal design of the health system and its organisation in health areas and zones".

Concerning tariffs, all MHOs sought to negotiate flat rates per type of service, in accordance with the national policy. Exceptional services or medicines not routinely covered were charged separately as fee-for-service. MHO leaders and medical advisers illustrated how providers interpreted the rule on tariffs to their advantage. Instead of an all-inclusive fee that covered consultation, diagnostic tests and medicines provided during the episode of illness, "the flat rate became the minimum rate charged, and any test or medicine was invoiced separately, significantly increasing the bill". For example, in a rural hospital where the agreed flat rate for a C-section was $60, the amount charged could more than triple when all separately charged exams, medications, or procedures were added. The effects of the agreed tariffs on practices is similar to that of covered services: "the list of agreed tariffs is used to determine what the MHO will pay"; "revision of invoices according to the contract is accepted, but does not change prescribing". Instead, "additional fees will be charged to the patient".

All MHOs required the preferential use of essential generic drugs, but most accepted the use of non-essential drugs if patients' conditions required it and the medical advisers approved of their prescription. Financial coverage was usually partial, for example 50% in MHOs of the CGAT network, with the aim to limit their use–the underlying assumption was that providers would want to avoid high expenses for their patient. In practice, prescribing generics was customary in most health centres and rural hospitals. In cities, on the other hand, prescribing non-generic drugs was commonly used to inflate the bill or respond to patient demand. One urban MHO, for example, counted every month about 300 prescriptions for non-generic medicines for an average 611 episodes. According to MHO managers, "prescribers convince patients that non-generic drugs are better, even when the MHO tells differently". Particularly, specialists could not be persuaded to prescribe generics: "They say we want to dictate how they should treat patients".

Concerning the clause on respect of national treatment guidelines, guidelines in the form of decision-making trees, were available and respected in most public health centres. Problems arose when the guidelines indicated that patients should be referred. According to health zone officials, "patients are often treated on-site to increase the health centre's income". In hospitals, the only national guidelines were those introduced by disease-specific programmes. For other diseases, some hospitals had designed their own guidelines in cooperation with international partners. Of importance, all interviewed medical doctors felt that "it is not the business of the MHO to propose treatment guidelines". All medical advisers acknowledged their limited say over prescribing: "The providers prescribe, and we pay services according to the contract". Also hospital managers and BDOM administrators testified of the difficulty to influence prescribing in their health facilities.

Thirdly, regarding the control of contract application by medical advisers, the verification process had mixed results. On the one hand they succeeded in drastically reducing the amounts to be paid by the MHOs. On the other hand, they found that they made little progress in improving healthcare quality and providers' responsiveness to patients' needs. Most medical advisers operated as part of the team at the technical support centres. Only MHOs with large memberships, i.e., MESP and MUSECCO, had their own medical adviser(s). Controlling the invoices issued by healthcare providers was their main role. To process the verification, they had access to all registers and patient medical records. The work involved checking whether invoiced services were provided, whether the MHO covered them, whether treatment guidelines and tariffs were respected, and whether prescribed diagnostic services or medicines were justified. The verification led to substantial savings for the MHOs. The medical advisors

estimated that about 95% of the monthly invoices were revised downwards. The gap between original and revised invoices could be substantial, up to 25% in some hospitals. Over time, MESP succeeded in significantly reducing overcharging. By 2016, services rejected for payment by the medical advisers represented less than 1% of healthcare expenses. They attributed this achievement to the fact that providers wanted to avoid the delay in payment caused by the verification process. Overall, medical advisors observed that their work did not have a durable effect on prescribing and billing practices and "needed to be vigorously repeated every month".

In summary, when discussing the effectiveness of strategic purchasing, issues of cost rather than quality of healthcare dominated respondents' concerns. For providers, the financial contribution of the MHO could promote quality when it improved medicine availability. However, many MHOs had growing financial difficulties and consequently paid irregularly or decreased their financial coverage of services. Others accumulated considerable arrears in payments owed to hospitals, which invariably influenced provider attitudes towards MHO members. For MHO managers, the contract positively influenced what the MHO paid for care provided. This influence remained, however, limited to the letter of the contract and did not fundamentally change provider behaviour. Providers in need of cash tended to work around the fixed tariffs and conditions. To cope with this problem, UMUSAC created its own health centre, which UMUSAC leaders found an adequate solution to "avoid overcharging, to ensure strict application of treatment flowcharts and rational prescribing". Several leaders of other MHOs shared the view that having their own health facilities that shared their values would serve their members best.

Finally, all respondents mentioned one fundamental factor that underlies the poor results of strategic purchasing as a means to improve care quality and affordability: "the absence of the health authorities". Providers distinctly expressed the opinion that imposing rules, treatment guidelines, and tariffs and controlling quality is the responsibility of the health authorities, not the MHOs. In the opinion of the interviewees, measures to promote rational prescribing can be effective only when the health authorities enforce them.

### M3—guarding over the long voice route: Link with more voiced groups or hold authorities accountable to regulate health services

Positive effects of linking with more voiced groups to hold providers accountable were illustrated by a respondent involved in launching the Nyantende MHO in 2001: "Healthcare providers will increase fees, overcharge, overprescribe when the MHO management is not strong. To be strong, real cooperation with local authorities, religious authorities, and school directors is necessary." This view guided the preparations and launching of the Nyantende MHO that soon became the largest MHO in South Kivu, with over 13,000 beneficiaries in 2013. Currently, building such cooperation may have become more difficult. Respondents in North and South Kivu pointed at a great distrust where matters of money are concerned as underlying factor explaining lack of cooperation. In the healthcare sector, they particularly emphasised the role of a history of free healthcare funded by external donors, and distrust about how these funds were used: "The great majority of the population, including healthcare providers and authorities, are convinced that the MHOs receive substantial external subsidies to pay for healthcare. When MHOs ask for people's contributions, they must have squandered the provided funds"; "When local administrative and health authorities understand that external funding is not there, they do not expect that anything can be achieved."

Furthermore, most organisations operated within their own network, with minimal links to other organisations. International non-governmental organisations and bilateral donors supported their specific programmes. Churches created their own MHO, with links to their own

healthcare system, schools, and community organisations. The network of MHOs in South Kivu, for example, "is integrated into the Catholic Church that has authority as one of the most effective organisations but is for the same reason resented by administrative authorities and other organisations". Although UMUSAC had good working relations with health authorities and providers, its main network was that of the Christian Workers Movement, nationally and internationally. MESP was spreading its own large network of MHOs over several provinces. The support centres CAMS, CGAT, and UMUSAC did stimulate cooperation and dialogue with and among the MHOs of their network, but this dynamic was essentially inward looking.

One notable exception breaks this prevailing isolationist attitude. To foster dialogue between MHOs and their healthcare providers, the CGAT organised regular meetings that brought together providers, MHOs, and local health authorities. Interviewees who participated found this "platform for dialogue" very useful for understanding the problems of others and moving towards solutions. Medical advisers in Kinshasa and Goma were confident that regular exchange would progressively influence provider attitudes. A similar platform was envisaged in Butembo where, after their involvement in MUSOSA's financial feasibility study, the District health authorities expressed interest in sustained cooperation to promote rational prescribing. CAMS in South Kivu lacked the financial resources to organise this type of interactions. Instead, the MHO leaders of Walungu and Nyantende participated in meetings of the health zone management committee where "problems were discussed, but will little effect on practices." Within the network of the Catholic Church, MUSECCO had helpful quarterly meetings with the BDOM coordination, who invited MHO leaders to the monthly providers' meetings when preoccupations needed discussion. The health authorities were not involved.

## M4—short voice route: Transforming the power imbalance at the provider–patient interface; increasing peoples' confidence to discuss directly with providers

No MHO encouraged its members to discuss issues of healthcare quality and affordability directly with the providers. MHO leaders shared the opinion that "providers do not listen to patients". However, other processes had been introduced to protect members in the provider–patient interaction, among which 1) the presence of an MHO delegate in the health facilities, 2) the existence of a feedback procedure to channel complaints, and 3) member control over MHO management to ensure that their priorities were implemented.

Firstly, the MESP in Kinshasa employed MHO delegates in the contracted health facilities who monitored whether members and providers respected the agreed-on procedures. Their presence proved effective in defending patient interests, improving patient care, and significantly reducing unnecessary expenditure. Managers of other MHOs also would have liked to test "whether appointing a permanent agent in the health facilities most used by members would improve patient care and reduce fraud".

Secondly, with respect to the feedback procedures, we distinguished several steps to assess their effectiveness: a) member knowledge about their rights and obligations; b) the functionality of feedback from members to MHO and from MHO to providers; and c) whether this changed the provider–patient relationship. Overall, members' understanding of their rights and obligations seemed to be related to their level of education and MHO longevity. Members of MUSECCO and MESP, the two teacher MHOs, were well informed, as were members of MUSOSA, Nyantende, and UMUSAC. Members of the younger MHOs of Walungu and the CGAT networks in Goma and Kinshasa had a poor understanding of the notion of rational

prescribing and tended to push for overconsumption. The feedback mechanism to report problems worked well in all MHOs, although feedback had little impact on provider behaviour in Goma and South Kivu.

Thirdly, regarding member control, all MHOs had the same organisational structure, which consisted of a general assembly of members with decision-making power, a board of directors in charge of translating decisions into strategies and supervising their execution, a control committee with the role of verifying the accounts, and a management team. The effectiveness of member participation and control also increased with education level and the longevity of the MHO. Members of MUSECCO and MUSOSA, for example, were very actively involved in decision-making, while in Lisanga and Walungu member control mostly consisted in approving by vote decisions made by their leaders.

To sum up, each of the three procedures introduced to transform power relations showed positive effects. However, their effectiveness in promoting care quality and affordability depended on other, often external factors that had more weight. For instance, MESP, with its considerable financial resources, political backing, professional staff, and large membership did promote change in care provision. Their influence did not, however, reach the larger hospitals "which do not need us". The voice of MUSECCO was heard through its privileged relations with the BDOM within the Catholic network. Other MHO leaders and support centre staff said that they could do little to influence care quality and affordability. They cited two main causes: insufficient financial means to perform the tasks needed to promote change, and their lack of authority in the discussion with providers. The technical support centres made progress in uniting MHOs in federations, thus strengthening their voice, but this progress had not yet transformed the power balance: "In our dealings with healthcare providers, the providers have the upper hand".

## Discussion

The framework we have used in this study distinguishes four mechanisms for MHO members to activate in influencing providers: (1) 'exit' or 'voting with the feet'; (2) 'co-producing a long voice route' or imposing rules through strategic purchasing; (3) 'guarding over the long voice route of accountability' or pressuring authorities to regulate and enforce regulations; and (4) 'strengthening the short voice route' by transforming the power imbalance at the provider–patient interface. All studied MHOs used these four mechanisms to improve healthcare provision. The results of our investigation, however, indicate that the outcomes of these processes are poor. Indeed, most healthcare providers do not recognise the authority of MHOs and their members to control quality and affordability of healthcare. The latter are firmly seen as the prerogative of the health authorities, even if these authorities only marginally take up this role. Under current circumstances, the power of MHOs in the DRC to enhance quality and affordability of healthcare remains unfortunately weak. These findings are consistent with previous study results highlighting the weak bargaining power of MHOs in their relationship with providers. Reasons evoked in this earlier work are their relative insignificance for hospital finances [23], a position of monopoly of providers in rural areas [24], and power imbalances [25,26].

Our findings contribute to deepen current insights on the influence of MHOs on care provision. When some positive influence is observed, it concerns nonclinical aspects such as patient reception, cleanliness, and sometimes greater respect for agreed tariffs, but not prescribing practices [27–32]. Yet, contracts between MHOs and healthcare providers almost always deal with clinical care quality and rational prescribing, specifying the same measures as those established for the MHOs we investigated [19]. In our study, we therefore looked in detail at the effectiveness of each of these measures to understand what worked, what did not, and why, using the analytical

framework of a similar study in Indian settings [20]. This framework allowed us to identify strengths and weaknesses of MHO interventions to improve care provision implemented in the DRC. However, data analysis brought forward the need to redefine three fundamental concepts that may contribute to further finetune the Michielsen framework, and more specifically three of the four mechanisms the framework proposes (M1, M2 and M4):

- It is necessary to disentangle the two complementary but distinct aspects of care quality and its affordability. Because the mission of MHOs is to facilitate access to quality care at affordable prices, both quality and cost and their relative importance need to be explicitly differentiated in M2 (strategic purchasing) to identify more specific interventions for change.

- The notions 'improving financial access' and 'decreasing expenses for healthcare' should also be clearly differentiated. Reducing healthcare expenses specifically refers to actions the MHO undertakes to incite cost-containment. Improving financial access is a wider notion that also refers to the effects of the insurance mechanism (prepayment and risk sharing) on accessibility. Contrary to Michielsen et al, we did not consider "Reduction of financial barriers via insurance" (M1) as a variable, but retained only those measures put into place to directly influence care provision.

- Transforming the power imbalance at the provider–patient interface (M4) should focus on the potential transformative power of the MHO as a member-based organisation rather than on the personal emancipation of patients in their interaction with providers. Two requirements suggested by Michielsen et al to make "direct negotiation with providers over care quality" possible are "no power imbalance at provider/patient interface" (M1) and "members are capable of evaluating both technical and interpersonal quality of care" (M4). Because knowledge asymmetry and therefore power imbalance are inherent to any provider–patient interface, we rather emphasised the role of the MHO, as representative of its members, to protect patients in their interaction with providers and to promote care quality on their behalf. Members, in turn, control the MHO management.

A worthwhile feature of our study is that it allows for some level of comparison between MHO schemes in DRC and India, despite the contextual differences. The same analytical approach was indeed used in both settings calling for a wider application of the original Michielsen framework. The findings of both studies in the DRC and Indian schemes show striking similarities. Both in India and the DRC, scarcity of health facilities offering quality care at affordable prices limited choice and therefore the effectiveness of the exit route (M1). The MHOs in the DRC engage in active strategic purchasing (M2), using the recommended strategies for promoting quality and cost containment, but as in the Indian schemes, are not successful in changing provider behaviour. Over-prescribing, overcharging, fraud, and poor patient reception persist. Accountability of providers to the MHOs is accepted regarding revision of invoices. Overall, however, providers accept quality control only from the health authorities or eventually from the coordination of their network, who, again as in India, are not always able or willing to execute this control. Linking with more voiced actors to hold providers accountable (M3) was therefore also not successful. In both countries, some schemes– and in the case of the DRC the MHO coordination bodies PNPMS and POMUCO–had strong links to policymakers at the national level and significant influence on social protection policies. Such links were missing at the local level in the DRC because of a general disinclination of organisations to cooperate with each other and because of poor involvement of administrative and health authorities.

Transforming the power imbalance at the provider–patient interface (M4) seemed for Michielsen et al the most promising way to improve care quality. Where social workers

accompanied members during hospitalisation, their presence had a substantial effect on provider attitudes towards patients. Similarly, in the DRC, the presence of MHO delegates in health facilities seemed to be the most effective measure to protect members and reduce invoicing for non-rendered services. On the other hand, the organisational structure of the MHO that gives members a greater say does not seem to empower MHOs in their relationship with providers. Other studies in African countries also found that the participatory dynamics of MHOs positively affected internal cohesion but not transformation of the provider–patient interface [27,33,34,35]. Creating their own health facility that shared their values was a satisfactory solution for UMUSAC, which other MHOs wished to emulate but lacked the financial resources to try out.

Should MHOs in the DRC lose faith in their capacity to improve access to quality healthcare? Our respondents agree that progress can be made only if health authorities play their role. Stating this fact is however not the solution. Someone needs to push for change, holding health authorities and providers to account. Notwithstanding the difficulties they encounter, MHOs still seem the best candidates for doing so. For the technical support centres, this aim implies investing in greater dialogue and cooperation with all actors involved in healthcare provision and development. The forum for dialogue created by the CGAT seems a promising first step.

Of note is that MESP stands out among the studied MHOs both in number of members and capacity to influence providers. This can largely be attributed to their financial situation secured by member contributions withheld from salaries and substantial government subsidies. This enabled the MHO to hire sufficient personnel, to develop adequate management structures and to implement planned strategies, which, in turn triggers a virtuous circle of gaining members' confidence, expanding and increasing their power.

## Limitations

A limitation of the present study is that it is based on secondary qualitative data collected in an earlier investigation on MHOs conducted 3 years earlier, i.e. in 2016. In the meantime, policies concerning MHOs may have evolved: see for instance the promulgation in February 2017 of the law on the principles of MHOs. We believe however that this limitation is mitigated, and is unlikely to have affected the quality of our paper, given the fact that three of the authors of the present paper (MPW, FKW and BC) were involved in the 2016 study and have, since then, closely monitored MHO policies in the DRC. Another limitation is the fact that MHO members themselves were not interviewed for this study. This could possibly have led to more insight in the mechanisms for MHO members to influence providers, especially M1 and M4. We nevertheless reported their opinions or behaviours when all respondents who mentioned them had the same interpretation.

## Conclusions

The premise that MHOs can improve care quality and affordability by their collective action is limited by the weight of contextual factors. In the DRC, measures enacted by MHOs to promote change encounter obstacles that MHOs cannot overcome on their own. Active cooperation with local health authorities is needed to reform the professional culture of healthcare providers. Three measures implemented by individual MHOs show potential to instigate change and should be explored further. The presence of an MHO delegate within the health facility had immediate effects on observance of some of the agreed procedures. Operating their own health facility granted MHOs direct control over service quality. A formal forum for dialogue, where MHOs, providers, and health authorities meet, showed potential for gradual

change in attitudes. Applying the framework for analysis in less challenging environments could provide more evidence to suggest alternative solutions.

## Supporting information

**S1 File.**
(DOCX)

**S2 File. Summary of results per MHO.**
(DOCX)

## Acknowledgments

This study is based on material from previous research, commissioned by POMUCO in 2016. Prof. Sylvain Shomba from the University of Kinshasa participated in the data collection for the material used for this study.

## Author Contributions

**Conceptualization:** Bart Criel, Maria-Pia Waelkens.

**Data curation:** Maria-Pia Waelkens.

**Formal analysis:** Maria-Pia Waelkens, Yves Coppieters, Samia Laokri.

**Funding acquisition:** Bart Criel.

**Investigation:** Maria-Pia Waelkens, Fulbert Kwilu Nappa.

**Methodology:** Bart Criel, Maria-Pia Waelkens, Samia Laokri.

**Project administration:** Bart Criel.

**Supervision:** Bart Criel.

**Writing – original draft:** Maria-Pia Waelkens.

**Writing – review & editing:** Bart Criel, Fulbert Kwilu Nappa, Yves Coppieters, Samia Laokri.

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
