## [Decision Letter · Decision Letter 0]

20 Jan 2020

PONE-D-19-28775

Can mutual health organisations influence the quality and the affordability of healthcare provision? The case of the Democratic Republic of Congo

PLOS ONE

Dear Dr. Criel,

Thank you for submitting your manuscript to PLOS ONE. After careful consideration, we feel that it has merit but does not fully meet PLOS ONE’s publication criteria as it currently stands. Therefore, we invite you to submit a revised version of the manuscript that addresses the points raised during the review process.

In order to provide a more complete information to our readers on the topic, we would like to emphasize the importance to cross referencing very recent material on the same topic published in "PLoS ONE ". Therefore, it would be highly appreciated if you would check the contents published in the last two years of "PLoS ONE" (https://journals.plos.org/plosone/) and add all material relevant to your article to the reference list.

We would appreciate receiving your revised manuscript by Mar 05 2020 11:59PM. To enhance the reproducibility of your results, we recommend that if applicable you deposit your laboratory protocols in protocols.io, where a protocol can be assigned its own identifier (DOI) such that it can be cited independently in the future. For instructions see: http://journals.plos.org/plosone/s/submission-guidelines#loc-laboratory-protocols

We look forward to receiving your revised manuscript.

Kind regards,

Wen-Jun Tu

Academic Editor

PLOS ONE

Journal Requirements:

2. Please address the following:

- Please ensure you have thoroughly discussed any potential limitations of this study within the Discussion section.

- Please provide additional details regarding participant consent. In the ethics statement in the Methods and online submission information, please ensure that you have specified how verbal consent was documented and witnessed.

- Please include additional information regarding the interview guide used in the study and ensure that you have provided sufficient details that others could replicate the analyses. For instance, if you developed a guide as part of this study and it is not under a copyright more restrictive than CC-BY, please include a copy, in both the original language and English, as Supporting Information. Please also include details of any pilot testing of the guide and its development.

Thank you for your attention to our queries.

This study is based on material from previous research, commissioned by POMUCO in 2016, and supported by the Belgian Directorate-general Development Cooperation and Humanitarian Aid in the context of the programme MASMUT (Micro-assurance santé/Mutuelles de santé).

Reviewers' comments:

Reviewer's Responses to Questions

**Comments to the Author**

1. Is the manuscript technically sound, and do the data support the conclusions?

Reviewer #1: Yes

2. Has the statistical analysis been performed appropriately and rigorously? 

Reviewer #1: N/A

3. Have the authors made all data underlying the findings in their manuscript fully available?

Reviewer #1: Yes

4. Is the manuscript presented in an intelligible fashion and written in standard English?

Reviewer #1: Yes

5. Review Comments to the Author

Reviewer #1: This article illustrates the influence of MHO in the healthcare system of CDR from different perspectives and gives an interesting insight into the challenges of providing quality care. I am sure other countries can relate and profit from the results. The qualitative approach allows to gather important information on the current situation and the authors propose further improvements. However, I think the article would profit from a better description of the state of research regarding MHO, a clear research question and a better structure.

Introduction:

I suggest first to describe the organization of the healthcare system in DRC (and the challenges) and then to explain why MHO were developed and how.

Does any research exist on MHO in DRC or other countries? Please provide information about previous research.

P 4, line 102-103: Please provide a reference for the online articles.

P 7-8. Please state the goal of this study more clearly.

P 7, line 170: I suggest mentioning the framework for analysis developed by Michielsen et al (2011) in the method section or mentioning it earlier in the introduction in the context of describing the research done in India.

Methods

P 8, line 190-194 Please give more information on why the data was re-analysed.

P 8, line 104: Who were the other key informants? Please give some information.

P 10, 233-235. Why do themes needed to be added? Why do are only a part of the changes explained in the method section and others in the discussion section?

Discussion

I suggest starting with a summary of your results.

P 26: the suggested redefinition of the three fundamental concepts appears to be an important benefit from your study, because it can improve the MHO concept and further research. I think it useful to emphasize more on these suggestions.

Please give more detailed limitations e.g. use of interviews which can lead to socially desired answers and reflect on sample selection. Maybe also reflect on the use of the framework by Michielsen.

6. PLOS authors have the option to publish the peer review history of their article (what does this mean?). If published, this will include your full peer review and any attached files.

Reviewer #1: Yes: Oriana Handtke

---

## [Author Response · Author response to Decision Letter 0]

28 Feb 2020

Antwerp 13th February 2020,

To the editorial board of PLOS ONE

Re: PONE-D-19-28775 Can mutual health organisations influence the quality and the affordability of healthcare provision? The case of the Democratic Republic of Congo

Dear Madam, Sir,

Thank you very much for sharing the results of the review process of our above-mentioned manuscript (mail of Wen-Jun Tu, academic editor PLOS ONE to us on Monday 20th of January 2020). We are grateful and believe these comments are useful to enhance the quality and readability of our manuscript.

We will address the editorial issues/journal requirements raised point by point.

1. We have adapted the manuscript to PLOS ONE’s style requirements

2.1. We have further developed the study limitations: see page 31/35, text highlighted in blue, lines 704-709

2.2. We have provided more information regarding participant consent, and more specifically the verbal consent: see page 10/35, text highlighted in blue, lines 213-226

2.3. We have prepared a specific word file (French and English: ‘Interview guides.docx’) presenting the structuring of the various questionnaires used in the study. We will introduce a copy of it as ‘Supporting Information’. 

3. We have obtained an ORCID iD: https://orcid.org/0000-0002-1452-0088

4.1. We have adapted the text in the ‘Acknowledgments Section’ of the manuscript, and more specifically removed the funding-related text: see page 32/35, lines 727-729

4.2. We have adapted the Funding Statement section of the online submission form accordingly

Review comments to the author formulated by the reviewer.

Reviewer #1: This article illustrates the influence of MHO in the healthcare system of CDR from different perspectives and gives an interesting insight into the challenges of providing quality care. I am sure other countries can relate and profit from the results. The qualitative approach allows to gather important information on the current situation and the authors propose further improvements. However, I think the article would profit from a better description of the state of research regarding MHO, a clear research question and a better structure.

Introduction:

I suggest first to describe the organization of the healthcare system in DRC (and the challenges) and then to explain why MHO were developed and how. Does any research exist on MHO in DRC or other countries? Please provide information about previous research.

Our answer: we now start the background section with a description of the organization of the healthcare system in DRC. See pages 3/35 and 4/45, lines 83-99. This makes well the transition with the paragraph starting at line 101 wherein we present the rationale for the DRC to opt for health insurance, and more specifically for Mutual Health Organisations. On page 5/35, lines 117-124, we have introduced a section on previous research in the DRC on MHOs, with a reference to the background paper on MHOs drafted for the World Health Report 2010. 

P 4, line 102-103: Please provide a reference for the online articles.

P 7-8. Please state the goal of this study more clearly.

Our answer: we clarified the goal of the study on pages 7/35 and 8/35, lines 173-179

P 7, line 170: I suggest mentioning the framework for analysis developed by Michielsen et al (2011) in the method section or mentioning it earlier in the introduction in the context of describing the research done in India. 

Our answer: we have mentioned the Michielsen et al. framework already in the background section on page 8/35, lines 178-179. We further develop it in the methods section on pages 10/35 and 11/35, lines 233-245. 

Methods

P 8, line 190-194 Please give more information on why the data was re-analysed.

Our answer: we have done so in the methods section page 8/35, lines 187-190; but also in the abstract page 2/35, lines 55-58.

P 8, line 104: Who were the other key informants? Please give some information.

Our answer: we have addressed this on page 8/35, line 200.

P 10, 233-235. Why do themes needed to be added? Why do are only a part of the changes explained in the method section and others in the discussion section?

Our answer: we clarified this in the methods section on page 11/35, lines 256-259. We have removed this from the discussion section. 

Discussion

I suggest starting with a summary of your results.

Our answer: we do so now. See pages 27/35 and 28/35, lines 610-623? 

P 26: the suggested redefinition of the three fundamental concepts appears to be an important benefit from your study, because it can improve the MHO concept and further research. I think it useful to emphasize more on these suggestions.

Our answer: we have emphasized this on page 28/35 in lines 636-637.

Please give more detailed limitations e.g. use of interviews which can lead to socially desired answers and reflect on sample selection. Maybe also reflect on the use of the framework by Michielsen.

Our answer: as already highlighted above, we further developed the section on study limitations on page10/35, lines 704-709.

---

## [Decision Letter · Decision Letter 1]

13 Mar 2020

PONE-D-19-28775R1

Can mutual health organisations influence the quality and the affordability of healthcare provision? The case of the Democratic Republic of Congo

PLOS ONE

Dear Dr. Criel,

Thank you for submitting your manuscript to PLOS ONE. After careful consideration, we feel that it has merit but does not fully meet PLOS ONE’s publication criteria as it currently stands. Therefore, we invite you to submit a revised version of the manuscript that addresses the points raised during the review process.

We would appreciate receiving your revised manuscript by Apr 27 2020 11:59PM. To enhance the reproducibility of your results, we recommend that if applicable you deposit your laboratory protocols in protocols.io, where a protocol can be assigned its own identifier (DOI) such that it can be cited independently in the future. For instructions see: http://journals.plos.org/plosone/s/submission-guidelines#loc-laboratory-protocols

We look forward to receiving your revised manuscript.

Kind regards,

Wen-Jun Tu

Academic Editor

PLOS ONE

Reviewers' comments:

Reviewer's Responses to Questions

**Comments to the Author**

1. If the authors have adequately addressed your comments raised in a previous round of review and you feel that this manuscript is now acceptable for publication, you may indicate that here to bypass the “Comments to the Author” section, enter your conflict of interest statement in the “Confidential to Editor” section, and submit your "Accept" recommendation.

Reviewer #1: All comments have been addressed

2. Is the manuscript technically sound, and do the data support the conclusions?

Reviewer #1: Yes

3. Has the statistical analysis been performed appropriately and rigorously? 

Reviewer #1: N/A

4. Have the authors made all data underlying the findings in their manuscript fully available?

Reviewer #1: Yes

5. Is the manuscript presented in an intelligible fashion and written in standard English?

Reviewer #1: Yes

6. Review Comments to the Author

Reviewer #1: Dear authors,

thank you for letting me review your manuscript a second time. I feel it has improved, especially the introduction. However I feel the discussion needs further improvement. Please find my comments below.

Introduction

P 3, 83. Please give the full name of DRC with DRC in parentheses at first mentioning.

P 3, 83. Please explain the primary healthcare strategy and/or give a reference.

P3, 93. Please provide a reference.

P3, 101. Please use the abbreviation of DRC.

Discussion

P28, 623 Please reformulate the first sentence or delete it.

P28-30 I suggest a reorganization of the discussion. Firstly, discuss your results in the context of previous study results and then secondly, emphasize on the discussion of the new findings of your study.

31, 707-709. How does this fact influence the quality of your paper?

P 31, 709-710. How might this limitation influence the results? Please give some information.

7. PLOS authors have the option to publish the peer review history of their article (what does this mean?). If published, this will include your full peer review and any attached files.

Reviewer #1: Yes: Oriana Handtke

---

## [Author Response · Author response to Decision Letter 1]

25 Mar 2020

Antwerp 23rd March 2020,

To the editorial board of PLOS ONE

Re: PONE-D-19-28775 Can mutual health organisations influence the quality and the affordability of healthcare provision? The case of the Democratic Republic of Congo

Dear Madam, Sir,

I acknowledge receipt of the mail of Wen-Jun Tu, academic editor of PLOS ONE, from Friday 13th of March 2020. 

I wish to systematically address the issues raised by the reviewer under point 6 of the email (Review Comments to the Author).

- On page 3 (line 83) the full name of DRC is given with the abbreviation DRC in parentheses

- On pages 3 & 4, the primary health care strategy of DRC is explained. See lines 84-87 on page 3, and lines 93-100 on page 4. In addition 4 new references are provided to substantiate this: 3 on page 3 and 1 on page 4.

- On page 4, line 104 we have given a fifth new reference backing up the statement on the domestic underfunding and aid dependency of the Congolese health system.

- On page 4, line 113, we have removed ‘Democratic Republic of Congo’ and used the abbreviation DRC instead. 

- In the discussion on page 28 (line 635) we have deleted the sentence ‘This is not entirely surprising’. 

- We have attempted to reorganise the discussion as follows: i) making a better link between the summary presentation of the results at the start of the discussion with findings coming from previous studies (see lines 633-638); ii) pursuing the discussion from line 640 on with writing that ‘Our findings contribute to deepen current insights on the influence of MHOs on care provision’ (see lines 640-641); and iii) highlighting at the bottom of page 29, lines 674-678, the original (methodological) aspect of our study. 

- Section on limitations: see lines 724-730. We have tried to better explain in what way the involvement of three co-authors of the current study in the earlier (2016) POMUCO study may or not have affected the quality of the paper (lines 724-727). Finally, we highlight how a closer involvement of MHO members in the current study could have contributed to strengthen the results (lines 729-730). 

I hope we have adequately addressed the various comments of the editorial board, and hope that our paper can now be accepted for publication in PLOS ONE.

Best wishes,

Bart Criel, MD, DTM&H, MSc, PhD

Professor, Department of Public Health

Institute of Tropical Medicine

Nationalestraat, 155, B-2000 Antwerp

bcriel@itg.be Tel (landline): +32 (0)3 2476293

---

## [Decision Letter · Decision Letter 2]

30 Mar 2020

Can mutual health organisations influence the quality and the affordability of healthcare provision? The case of the Democratic Republic of Congo

PONE-D-19-28775R2

Dear Dr. Criel,

We are pleased to inform you that your manuscript has been judged scientifically suitable for publication and will be formally accepted for publication once it complies with all outstanding technical requirements.

With kind regards,

Wen-Jun Tu

Academic Editor

PLOS ONE

Additional Editor Comments (optional):

Reviewers' comments:

Reviewer's Responses to Questions

**Comments to the Author**

1. If the authors have adequately addressed your comments raised in a previous round of review and you feel that this manuscript is now acceptable for publication, you may indicate that here to bypass the “Comments to the Author” section, enter your conflict of interest statement in the “Confidential to Editor” section, and submit your "Accept" recommendation.

Reviewer #1: All comments have been addressed

2. Is the manuscript technically sound, and do the data support the conclusions?

Reviewer #1: Yes

3. Has the statistical analysis been performed appropriately and rigorously? 

Reviewer #1: N/A

4. Have the authors made all data underlying the findings in their manuscript fully available?

Reviewer #1: Yes

5. Is the manuscript presented in an intelligible fashion and written in standard English?

Reviewer #1: Yes

6. Review Comments to the Author

Reviewer #1: Dear authors,

thank you for letting me review this article. You have addressed all my comments. I have no more comments.

7. PLOS authors have the option to publish the peer review history of their article (what does this mean?). If published, this will include your full peer review and any attached files.

Reviewer #1: No

---

## [Editor Report · Acceptance letter]

1 Apr 2020

PONE-D-19-28775R2 

Can mutual health organisations influence the quality and the affordability of healthcare provision? The case of the Democratic Republic of Congo 

Dear Dr. Criel:

I am pleased to inform you that your manuscript has been deemed suitable for publication in PLOS ONE. Congratulations! Your manuscript is now with our production department. 

With kind regards,

on behalf of

Dr. Wen-Jun Tu 

Academic Editor

PLOS ONE